# Attributes of Process Maturity of Public Administration Units in Poland

**Krzysztof Krukowski \*** 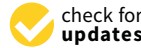 **and Magdalena Raczyńska**

The Faculty of Economics, University of Warmia and Mazury, 10-719 Olsztyn, Poland;
magda.raczynska@uwm.edu.pl
**\*** Correspondence: kkruk@uwm.edu.pl

**Abstract:** Process management is a concept that is used in public administration units in Poland to an increasing extent. Implementing this concept in public organizations, in line with the assumptions of New Public Management, is directed, among others, to increase their efficiency. The purpose of the research presented in the article was to identify the attributes describing process maturity of the community offices of urban type in Poland and to assess the interdependence of the attributes. In order to achieve the goal, an authors' questionnaire was used. Also, an attempt was made to create a process maturity model dedicated to the community offices. As a result of the conducted research, it was noticed that most of the examined entities use, at least, some elements (attributes) of process management. However, they are used at different levels by individual community offices.

**Keywords:** process management; process maturity; process maturity models; public administration units; community office

## 1. Introduction

Process management (process approach) is a concept that has been gaining popularity since 1980s. According to M. Weske, process management includes other concepts, methods, and techniques supporting design of processes, their administration, configuration, implementation, and analysis. Moreover, the basis of process management is a clear representation of processes along with their activities and implementation restrictions between them (Weske 2007, p. 5). The concept was initially used only in business organizations. The results of its application in these organizations, manifesting itself, among others by increasing their efficiency, encouraged managers of public organizations (public administration units) to implement this concept also in the entities they manage (1990s) (Houy et al. 2010, p. 627). These activities found support in the assumptions of the New Public Management concept, especially in its model based on the "trend of excellence"—one of its main objectives was to establish the need to implement in public organizations the solutions, which were originally used in business organizations (Hood 1991, p. 3).

The level (degree) of process management use in organizations is defined as their process maturity. It can be measured using the process maturity models. Determining the degree of process management implementation and development in a given organization allows not only to assess its advancement in the use of process-based solutions, but also to indicate actions aimed at raising the level of its process maturity. While public administration units, especially those of a local government nature[1], play a significant role in meeting the needs of citizens, the focus on continuous improvement of their

---

[1] The entities with the help of which the tasks of local government units are performed are, among others the community offices of urban type.

performance, including through the introduction and development of the process approach in these entities, may be considered a key activity.

The use of process management in public administration units is described in the literature rather in the context of the conditions for its implementation and development in these entities, or the degree of computerization of the processes that they execute (e-government)[2]. Publications regarding specifically the identification of attributes that describe process maturity of public organizations or aimed at creating process maturity models dedicated to these entities are still rare.

Bearing in mind the current state of knowledge on process maturity of public administration units and the impact of the application of the discussed concept on the efficiency of their functioning, the purpose of the research presented in this study is to identify the attributes describing process maturity of the community offices of urban type (community offices) in Poland and assessing their interdependence.

## 2. Theoretical Background

An important area of research on the use of process management in public organizations is to identify the attributes describing the level of their process maturity. This is due to the fact that introduction of process approach in organizations is characterized by going through certain stages, i.e., levels of process maturity (de Bruin and Rosemann 2007, pp. 642–53). If we assume that the maturity includes competences, abilities, level of advancement of the selected area of an organization, based on a, more or less, extensive set of attributes (criteria), process maturity can be defined as the degree of advancement of implementation of process approach in an organization. A basic assumption of the concept of process maturity is that mature organizations perform activities systematically, and non-mature organizations achieve their results through a one-off effort, using methods that are created by them in a, more or less, spontaneous way (Krukowski 2016, pp. 148–49). The specification of high and low process maturity of organizations was developed by M. Rosemann and T. de Bruin (Rosemann and de Bruin 2005). It is worth noting that the authors did not define the term "process mature organization" and "process immature organization" but focused on the attributes that describe the stages of process maturity, which are: Initial state, process definition, process repeatability, process management, and process optimization. According to the same authors, process maturity can be defined as the combination of "reach" and "fluency" of an organization (Rosemann and de Bruin 2005). The "reach" refers to an organization's ability and level of the process management principles' implementation, while "fluency" measures the quality and effectiveness of implementation of processes in an organization. To assess the maturity of an organization using these factors, the authors built a model[3] containing the set of criteria, that help to identify organizations with low and high process maturity. The level of maturity is demonstrated in this model: By the number of processes managed by the organization; by the involvement of employees and managers performing activities related to process management; and by the use of process management tools. And to assess fluency in the implementation of the process approach, we can use: Reactions to problems and initiatives related to implementation of process management; frequency of activities and initiatives; and also suitability of tools, resources and practices in process management.

Except from the above presented model, in the area of process management, many other maturity models can be identified. According to a list made by B. Champlin, the President of the Association of

---

2　e.g., Tregear and Jenkins (2007).
3　Most of the process maturity models are based on the evolutionary development of an organization. They describe the stages of an organization's process development and the path to reach a given stage. Each stage must have specific features and they have to be in logical relation to the features of subsequent stages. It is also important that they are created on the basis of some, more or less, extensive attributes of process maturity, which are defined as specific, measurable, and independent elements, reflecting the basic and separate characteristics of process management. These criteria, allowing to determine the desired or logical path of an organization's process evaluation, create a predictable pattern of organizational evolution and changes taking place in an organization. See: Röglinger et al. (2012), p. 330.

Business Process Management Professionals (ABPMP), there are about 150 different types of process maturity models (Spanyi 2004, p. 1). Most of them are based on the assumptions formulated by G. Rummler and A. Brache, which follow the premises of achieving and measuring organizations' effectiveness (Rummler and Brache 2000, pp. 74–94). J. Pöppelbuß and M. Röglinger identify two types of maturity models (Pöppelbuß and Röglinger 2011), in which many detailed models exist. The first type concerns process maturity related to the state of individual processes in an organization, and in particular to whether they are managed, documented, and performed. The second concerns the process maturity of an organization as a whole. An example of the first type of models could be the Capability Maturity Model Integration (CMMI), developed at the Software Engineering Institute, which includes five levels of process maturity, ranging from a chaotic approach to the stages that constantly improve the process. The CMMI model was an inspiration to create many other process maturity models like the Business Process Maturity Model (BPMM)[4], which strictly adheres to the principles of the Process Maturity Framework (Humphrey 1987, p. 3). The second type of models refers to the maturity of the process management based on the possibility of introducing the process approach in an organization as a whole (Röglinger et al. 2012, pp. 328–46). These models are designed to provide a holistic assessment of all the fields in organizations related to managing their processes. They usually include many aspects, such as management, methods and tools, and organizational culture (Rohloff 2009, p. 133), but also process efficiency as a separate dimension (Hammer 2007, pp. 111–23). For example, according to the Gartner group model, organization's process maturity can reach six levels (Melenovsky and Sinur 2006, p. 6). Achieving the first level can be interpreted as information that a particular organization meets the minimum requirements to implement the discussed concept, and the highest level—that the organization uses the concept optimally. The attributes for assessing the level of process maturity in this model are: Organizational behavior, human resources, management, IT technologies, and applied methods and techniques. It should be emphasized that each of these determinants, according to the discussed model, is characteristic in each of the stages of process maturity of an organization (Kerremans 2008, pp. 7–14). On a similar basis, the holistic process maturity models were built by (Fisher 2004)[5] or (de Boer et al. 2015)[6].

A special type of process maturity models is that, dedicated to public organizations. Creating models of process maturity dedicated to public organizations makes sense, because often the determinants of their functioning are different than those of business organizations (Ramos et al. 2019, p. 190; Szumowski and Cyfert 2018, pp. 16–17). In the literature, especially the models of process maturity of public administration units without a foundation on the CMMI model can be found (Zwicker et al. 2010, pp. 369–95). They do not focus on the aspect of public management implementation in public organizations, but on the assessment and improvement of processes related specifically to e-government. These models can be divided into models developed on the basis of theory analysis and on the basis of observation of practice. In most of these models, process maturity is used as one of the criteria, but they mainly serve as tools for assessing and improving provision of public services by electronic means. Each of them includes several levels of maturity of access to these services via the Internet. For example, one of the models developed by P. Gottschalk identifies five levels of public organization maturity due to the electronic provision of services (Gottschalk 2009, pp. 75–81), and in the model proposed by K. Layne and J. Lee, there are four levels of e-services maturity. These are: "Cataloging", "Transactions" or "One-way interaction", "Vertical integration", and "Horizontal integration", assuming that the transition from one level to the next is associated with the increasing complexity of performed operations and their integration (Layne and Lee 2001, pp. 122–36). Also, the "48-h promise" model proposed by J. Zwicker, P. Fettke and P. Loos is worth mentioning[7].

---

[4]　See: Object Management Group (2008).
[5]　See: Fisher (2004).
[6]　See: de Boer et al. (2015).
[7]　See: Zwicker et al. (2010).

The models of process maturity of public organizations refer mainly to the one-dimensional linear models, i.e., those in which levels of maturity are distinguished. However, the essence of all the process maturity models is the use of attributes describing the level of process management and thus allowing to indicate the level of advancement of process solutions used in organizations. And the basis for assessing that level is to identify the attributes describing organizations' process maturity.

## 3. Methods

The subjective scope of research presented in this study included Polish community offices of urban type[8]. The sample for research was selected with the use of a non-random selection method, i.e., a non-probabilistic one. Within this method, the judgmental sampling technique was applied. The entities were selected on the basis of the objective set (Churchill 2002, p. 500), i.e., those community offices were chosen thanks to which information on the use of process management could be obtained. Thus, it was established that the selection of community offices representing cities with a population of over 20,000 inhabitants allows for this objective to be achieved (269 community offices[9]. While the number of community inhabitants translates into the number of community office's workers (Flieger 2012, p. 226), it was assumed that the surveyed entities are large enough for different process solutions to be applied in them[10].

In order to identify the attributes describing process maturity of the researched community offices, the authors' questionnaire was used. In the questionnaire the attributes were divided into four groups: (1) identification and improvement of processes occurring in community offices, (2) involvement of employees and managers involved in processes management, (3) deviations in process implementation, and (4) using tools to monitor process management.

The questionnaire was addressed to the secretary of the community office (city secretary). The choice of the city secretary as the respondent was supported by the fact that the objective of the person holding this position is to ensure proper organization of the community office's work and implementation of the policy in the area of human resources management. This requires that the city secretary possess a broad knowledge of the functioning of the community office, the subject of activity of its individual units and the functioning normative acts.

The research was conducted using the computer assisted telephone interview (CATI) method, i.e., a computer-aided telephone interview, supplemented with a questionnaire sent by e-mail at the respondent's request. As a result of the conducted survey, 174 valid questionnaires were obtained from the community offices (64.7% of the chosen sample). Such a high response rate may be related to the fact that in Poland public information is "open" and public organizations are very disciplined in participating in various research. In addition, the research was commissioned to a company specialized in conducting CATI research, which also affected the final number of responses received.

In order to examine the stochastic independence between the individual attributes of process maturity, the chi-square test of independence ($\chi^2$), also known as Pearson's chi-squared test, was applied (Aczel 2000, pp. 757–58). Following hypotheses were put forward:

**Hypothesis 1 (H1).** *Two attributes of the community offices' process maturity are mutually independent.*

**Hypothesis 2 (H2).** *Two attributes of the community offices' process maturity are not mutually independent.*

---

[8]　There are, in total, 302 community offices of urban type in Poland. See Główny Urząd Statystyczny (2018): https://stat.gov.pl/statystyka-regionalna/jednostki-terytorialne/podzial-administracyjny-polski/ (accessed on 10 September 2019).

[9]　The chosen sample accounted for 100% of the community offices of urban type in Poland, meeting the condition concerning 20,000 inhabitants of the commune.

[10]　The relationship between the size of an organization and the use of process-based solutions was indicated by e.g., R. Dijkman, S. V. Lammers and A. de Jong. See: Dijkman et al. (2016).

In further analysis, Kendall's $\tau$ test was applied. This method was used to examine the power and the direction of the existing dependence of the community offices' maturity attributes.

In both cases the level of significance of $\alpha = 0.050$ was used in the analysis.

## 4. Results

### 4.1. The Attributes of Process Maturity of the Researched Community Offices

The basic parameter considered in the assessment of process maturity of the researched community offices was the number of identified processes occurring in a given entity, because that element indicates the possibilities of applying process solutions in it.

In 34 studied community offices, no processes were identified (20%), which may indicate that process solutions were not used by these entities or, at least, that the respondents were not aware that the processes are actually performed in their community office. Therefore, these entities were omitted in further analyzes. In the case of 22% community offices, all their performed processes were identified (Figure 1). And most often, the studied entities identified over 60.0% of processes (29% of the community offices).

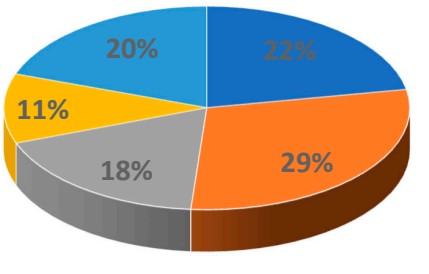

- ■ % of identified processes in community offices - 100%
- ■ % of identified processes in community offices >60%
- ■ % of identified processes in community offices 40%-60%
- ■ % of identified processes in community offices <40%
- ■ % of identified processes in community offices - 0%

**Figure 1.** Percentage of identified processes in the surveyed community offices (n = 174). Source: Own work based on research results.

The second analyzed distinguishing feature of the process maturity of the studied community offices was the determination of the number of plans to improve the identified processes (Table 1). It is an element that proves the community office's high involvement in the process-based development. Only in the case of 9.3% of the community offices, all identified processes had a developed improvement plan. Thus, a small number of the surveyed entities in this group proves the low level of use of tools to improve processes. In approximately 31.4% of community offices, such plans were prepared in the case of 40.0–60.0% of processes. Also, in the case of this determinant, there was a percentage of studied entities in which the identified processes did not have any improvement plans.

**Table 1.** The level of process optimization in the surveyed community offices (n = 140).

| Specification | % of Processes in Community Offices | | | | |
|---|---|---|---|---|---|
| | 100 | >60 | 40–60 | <40 | 0 |
| % of developed plans for improving processes | 9.3 | 29.3 | 31.4 | 21.4 | 8.6 |
| % of optimized processes | 7.9 | 29.3 | 37.1 | 17.1 | 8.6 |

Source: Own work based on research results.

In 29.3% of the surveyed entities, over 60% of identified processes were also optimized, but only 7.9% of the community offices indicated optimization of all processes. This can prove low process maturity, because improvement measures are characteristic for the higher stages of process maturity.

Another monitored factor when assessing process maturity of the community offices, was the monitoring of the execution of processes. One of the basic attributes of processes that is measured are costs. They cover all costs related to the implementation of activities that make up a given process. In Polish community offices, it is an index difficult to measure and often treated as a fixed cost. However, without knowing the cost of the process, the possibilities of its improvement are limited. Based on the results of the research, it can be concluded that the surveyed entities that had identified processes were also trying to monitor their costs. Only in the case of 5.7% of the community offices, these costs were not monitored (Table 2). In 30.0% of the surveyed entities, the costs of all processes were always monitored, and in the case of 34.3%—very often. In addition to the level of cost monitoring, another important attribute of process assessment is the monitoring of the time of their execution, i.e., the total time of performing particular activities in the process (process duration). This attribute indicates the level of organizing the procedures used in a community office, the used technologies and the employees' qualifications. Time, as an element of process monitoring, was used for all processes by 12.1% of the community offices. Only in 1.4% of them, this indicator was not considered in the process monitoring. Another attribute of process assessment is their quality, which can be understood traditionally—as a conditioned number of errors and related re-execution of the work or as customer satisfaction. In the case of quality, only about 7.9% of the surveyed community offices monitored all processes and in 41.4% of them, the quality was monitored very often.

**Table 2.** Community offices in which process characteristics are being monitored (n = 140).

| Specification | Frequency of Appearance (% of City Offices) | | | | |
|---|---|---|---|---|---|
| | **Always** | **Very often** | **Often** | **Occasionally** | **Never** |
| Monitoring of process costs | 30.0 | 34.3 | 16.4 | 13.6 | 5.7 |
| Monitoring of process quality | 7.9 | 41.4 | 32.1 | 13.6 | 5.0 |
| Monitoring of process duration | 12.1 | 28.6 | 42.9 | 15.0 | 1.4 |
| Monitoring of process recipients' level of satisfaction | 14.3 | 17.1 | 34.3 | 20.0 | 14.3 |

Source: Own work based on research results.

When monitoring processes, also the degree of satisfaction of their clients is taken into consideration by the authorities. Interestingly, in almost every fifth community office, the respondents did not have knowledge on this subject, and only in 14.3% of the surveyed entities the degree of process customer satisfaction was an attribute of evaluation of processes considered always when the processes were evaluated. To assess customer satisfaction, in 37.6% of the community offices surveys were used and in 25%—these were the solutions related to the common assessment framework (CAF) methodology.

The introduction of solutions based on processes requires the authorities to engage managers and regular employees. According to the respondents, the success of implementing process management depends on the total involvement of employees and managers only in about 15.0% of the community offices (Table 3).

The surveyed city secretaries also pointed out that they did not have knowledge about the impact of the community office's staff on the implementation of processes. However, in over 80.0% of the surveyed entities, the implementation of processes required the involvement of employees at various levels. Also, only in 15.4% of them there was a clear division of responsibilities in the execution of all processes. This is related to the fact that in community offices there are no solutions typical for process structures and the processes are implemented mostly within one organizational unit. This causes the responsibility for the execution of processes to be passed on to a given functional manager. What is

more, nearly 20.0% of the respondents did not have any knowledge about the division of responsibility in the execution of processes.

**Table 3.** The role of employees in the implementation of process management (n = 174).

| Specification | Frequency of Appearance (% of City Halls) | | | | | No Knowledge on the Subject |
| --- | --- | --- | --- | --- | --- | --- |
| | Always | Very often | Often | Occasionally | Never | |
| The success of implementing process management depends on the determination of employees | 13.7 | 25.1 | 33.1 | 12.6 | 0.6 | 14.9 |
| The success of implementing process management depends on the determination of managers | 14.9 | 23.4 | 21.7 | 18.9 | 2.9 | 18.3 |
| There is a clear division of responsibilities in the execution of processes | 15.4 | 24.0 | 19.4 | 20.6 | 1.1 | 19.4 |

Source: Own work based on research results.

Among the factors affecting the appearance of difficulties in implementing process management or negative phenomena showing the immaturity of process organization, the respondents pointed to: Improvising process execution, ad-hoc response to deviations in the execution of processes, non-compliance with processes and exceeding the time, and costs of executing processes (Table 4).

**Table 4.** Factors showing deviations in the implementation of processes in the surveyed community offices (n = 140).

| Specification | Frequency of Appearance (% of Community Offices) | | | | |
| --- | --- | --- | --- | --- | --- |
| | Always | Very Often | Often | Occasionally | Never |
| The execution of processes is improvised by employees | 0.0 | 10.7 | 29.3 | 37.1 | 22.9 |
| There is an ad hoc response to emerging crises related to the execution of processes | 19.3 | 29.3 | 24.3 | 25.7 | 1.4 |
| The specified processes are not respected | 2.1 | 12.1 | 18.6 | 38.6 | 28.6 |
| Process execution time is exceeded | 2.1 | 8.6 | 28.6 | 46.4 | 19.3 |
| The cost of processes is exceeded | 2.1 | 10.7 | 17.9 | 33.6 | 35.7 |

Source: Own work based on research results.

Most often, in the case of deviations, there were ad-hoc responses to emerging crises in the execution of processes. Over 19.3% of respondents claimed that this way of responding to deviations in the execution of processes always occurs in their organization, and 29.3% stated that this happens very often. This may serve as evidence of the lack of implemented comprehensive solutions for process management in the surveyed community offices. Moreover, over 29.3% of them often improvised the execution of processes and exceeded the time of their execution (28.6%). However, most respondents indicated that various deviations occur sporadically. This should be considered a positive phenomenon, demonstrating the increase of process awareness in the surveyed entities.

*4.2. Relation between the Individual Attributes of Process Maturity of the Researched Community Offices*

The examination of the stochastic independence between the individual attributes of process maturity of the researched community offices proved that there is a significant statistical dependence between the attributes. The test probability was $p = 0.000$ for all the pairs of attributes, which determines the rejection of the null hypothesis (H1). The results of this analysis are the basis of stating that the attributes are significantly different from each other. And only the assessment of all of them enables one to correctly diagnose the level of process maturity of the community offices. The power and the direction of the existing dependence is presented in Table 5 (where $\tau > 0$ means the occurrence of positive correlation, $\tau < 0$ means the occurrence of a negative correlation between the examined features). In the case of the analyzed attributes, a positive correlation was found between each of the

pairs analyzed. In all the cases considered, the value of Kendall's coefficient τ has assumed values above zero, therefore, it can be assumed that there was a significant positive correlation between individual attributes, despite the fact that their absolute value was low. In the case of only one of the analyzed attribute pairs (the existence of clear division of responsibility in the execution of processes and exceeding the time of process execution), the Kendall coefficient τ was zero. This indicates the non-existence of a monotonic relationship between the parameters studied. The obtained Kendall coefficients τ mostly represent a significant "consistency" of rank ordering (increasing monotonic dependence), i.e., the growth of the independent variable corresponds to the increase of the dependent variable in the case of the analyzed attributes.

In the case of analysis of attribute pairs in which the Kendall coefficient τ is close to zero, the correlations are irrelevant. In the case of an attribute saying that there is a clear division of responsibility in the execution of processes, correlation with other attributes is irrelevant in three cases: "execution of processes is improvised by employees" ($p = 0.59$), "specified processes are not observed" ($p = 0.37$), and "the time of processes' execution is exceeded" ($p = 0.99$). A similar situation occurs with the attribute "specified processes are not observed" and "number of processes in which their costs are monitored" (Kendall coefficient $\tau = 0.01$, $p = 0.85$). The irrelevant dependencies between the attributes can also include those occurring between the exceeding the costs of processes and the attributes like: "there is an ad hoc response to emerging crises related to the execution of processes" ($p = 0.22$), "the success of process management implementation depends on the manager's determination" ($p = 0.49$), and "the success of process management implementation depends on the employees' determination" ($p = 0.88$). Demonstration of the lack of correlation between these attributes may be due to the fact that in the case of public administration, the costs of processes are calculated rarely or not at all. The highest value of the applied correlation coefficient was obtained in the case of the pair of the attributes: "number of the improved processes" and "number of plans elaborated to improve the processes" (Kendall coefficient $\tau = 0.60$, $p = 0.00$). The same level of this coefficient occurs for the attributes: "number of processes in which their time of execution is monitored" and "number of processes in which their quality is monitored". On this basis, it can be concluded that the community offices that monitor the process duration, also monitor their quality.

**Table 5.** Analysis of the dependence of process maturity attributes (n = 174) (α = 0.05).

| | Attributes | Number of Identified Processes | Clear Division of Responsibility in the Execution of Processes | Execution of Processes Is Improvised by Employees | Specified Processes Are Not Observed | There is an ad hoc Response to Emerging Crises Related to the Execution of Processes | The Success of Process Management Implementation Depends on the Manager's Determination | The success of Process Management Implementation Depends on the Employees' Determination | Number of Processes in Which Their Costs Are Monitored | The costs of Processes Are Exceeded | Number of Processes in Which Their Quality Is Monitored | Number of Processes in Which Their Time of Execution Is Monitored | The Time of Processes' Execution Is Exceeded | Community Offices in Which the Level of Clients' Satisfaction Is Monitored | Number of Plans Elaborated to Improve the Processes | Number of Improved Processes |
|---|---|---|---|---|---|---|---|---|---|---|---|---|---|---|---|---|
| τ Kendall's coefficients (p) | Number of identified processes | - | 128.1 (0.00) | 82.6 (0.00) | 97.8 (0.00) | 129.9 (0.00) | 105.1 (0.00) | 123.3 (0.00) | 92.7 (0.00) | 108.6 (0.00) | 137.4 (0.00) | 125.1 (0.00) | 93.1 (0.00) | 139.1 (0.00) | 144.2 (0.00) | 129.4 (0.00) |
| | Clear division of responsibility in the execution of processes | 0.42 (0.00) | - | 94.5 (0.00) | 99.9 (0.00) | 197.4 (0.00) | 200.9 (0.00) | 206.9 (0.00) | 130.4 (0.00) | 105.3 (0.00) | 152.2 (0.00) | 156.2 (0.00) | 122.2 (0.00) | 191.1 (0.00) | 87.8 (0.00) | 100.9 (0.00) |
| | Execution of processes is improvised by employees | 0.20 (0.00) | 0.03 (0.59) | - | 114.6 (0.00) | 116.7 (0.00) | 92.9 (0.00) | 129.7 (0.00) | 41.8 (0.00) | 53.6 (0.00) | 83.9 (0.00) | 62.4 (0.00) | 104.5 (0.00) | 107.3 (0.00) | 92.6 (0.00) | 77.8 (0.00) |
| | Specified processes are not observed | 0.21 (0.00) | 0.05 (0.37) | 0.49 (0.00) | - | 143.9 (0.00) | 101.6 (0.00) | 131.4 (0.00) | 92.7 (0.00) | 119.5 (0.00) | 117.5 (0.00) | 90.6 (0.00) | 135.7 (0.00) | 153.9 (0.00) | 75.2 (0.00) | 98.1 (0.00) |
| | There is an ad hoc response to emerging crises related to the execution of processes | 0.33 (0.00) | 0.44 (0.00) | 0.36 (0.00) | 0.28 (0.00) | - | 195.8 (0.00) | 141.7 (0.00) | 67.1 (0.00) | 78.9 (0.00) | 155.2 (0.00) | 129.8 (0.00) | 92.8 (0.00) | 97.3 (0.00) | 119.4 (0.00) | 87.6 (0.00) |
| | The success of process management implementation depends on the manager's determination | 0.27 (0.00) | 0.56 (0.00) | 0.07 (0.16) | 0.18 (0.00) | 0.53 (0.00) | - | 185.7 (0.00) | 69.3 (0.00) | 69.5 (0.00) | 169.9 (0.00) | 132.5 (0.00) | 101.8 (0.00) | 129.8 (0.00) | 114.4 (0.00) | 121.4 (0.00) |
| | The success of process management implementation depends on the employees' determination | 0.37 (0.00) | 0.58 (0.00) | 0.19 (0.00) | 0.18 (0.00) | 0.45 (0.00) | 0.59 (0.00) | - | 88.4 (0.00) | 89.7 (0.00) | 195.4 (0.00) | 131.7 (0.00) | 110.7 (0.00) | 196.5 (0.00) | 96.8 (0.00) | 171.7 (0.00) |
| | Number of processes in which their costs are monitored | 0.22 (0.00) | 0.39 (0.00) | 0.10 (0.04) | 0.01 (0.85) | 0.20 (0.00) | 0.32 (0.00) | 0.37 (0.00) | - | 152.6 (0.00) | 94.8 (0.00) | 117.8 (0.00) | 89.0 (0.00) | 142.9 (0.00) | 102.1 (0.00) | 115.7 (0.00) |
| | The costs of processes are exceeded | 0.04 (0.49) | 0.16 (0.00) | 0.32 (0.00) | 0.21 (0.00) | 0.06 (0.22) | 0.04 (0.49) | 0.01 (0.88) | 0.11 (0.03) | - | 93.8 (0.00) | 67.9 (0.00) | 137.9 (0.00) | 98.1 (0.00) | 64.1 (0.00) | 68.4 (0.00) |
| | Number of processes in which their quality is monitored | 0.44 (0.00) | 0.49 (0.00) | 0.24 (0.00) | 0.15 (0.00) | 0.47 (0.00) | 0.46 (0.00) | 0.62 (0.00) | 0.39 (0.00) | 0.15 (0.00) | - | 245.6 (0.00) | 67.9 (0.00) | 166.5 (0.00) | 139.7 (0.00) | 118.6 (0.00) |

Test of Independence of the Attributes of Process Maturity χ² (p)

**Table 5.** *Cont.*

| | | Test of Independence of the Attributes of Process Maturity $\chi^2$ (p) | | | | | | | | | | | | | | |
|---|---|---|---|---|---|---|---|---|---|---|---|---|---|---|---|---|
| | Attributes | Number of Identified Processes | Clear Division of Responsibility in the Execution of Processes | Execution of Processes Is Improvised by Employees | Specified Processes Are Not Observed | There is an ad hoc Response to Emerging Crises Related to the Execution of Processes | The Success of Process Management Implementation Depends on the Manager's Determination | The success of Process Management Implementation Depends on the Employees' Determination | Number of Processes in Which Their Costs Are Monitored | The costs of Processes Are Exceeded | Number of Processes in Which Their Quality Is Monitored | Number of Processes in Which Their Time of Execution Is Monitored | The Time of Processes' Execution Is Exceeded | Community Offices in Which the Level of Clients' Satisfaction Is Monitored | Number of Plans Elaborated to Improve the Processes | Number of Improved Processes |
| τ Kendall's coefficients (p) | Number of processes in which their time of execution is monitored | 0.49 (0.00) | 0.45 (0.00) | 0.11 (0.02) | 0.03 (0.54) | 0.32 (0.00) | 0.42 (0.00) | 0.46 (0.00) | 0.42 (0.00) | 0.13 (0.01) | 0.60 (0.00) | - | 101.9 (0.00) | 128.5 (0.00) | 98.8 (0.00) | 116.3 (0.00) |
| | The time of processes' execution is exceeded | 0.19 (0.00) | 0.00 (0.99) | 0.42 (0.00) | 0.49 (0.00) | 0.16 (0.00) | 0.20 (0.00) | 0.14 (0.01) | 0.09 (0.06) | 0.50 (0.00) | 0.08 (0.10) | 0.23 (0.00) | - | 120.3 (0.00) | 45.9 (0.00) | 89.87 (0.00) |
| | Community offices in which the level of clients' satisfaction is monitored | 0.48 (0.00) | 0.48 (0.00) | 0.28 (0.00) | 0.28 (0.00) | 0.25 (0.00) | 0.32 (0.00) | 0.43 (0.00) | 0.30 (0.00) | 0.22 (0.00) | 0.54 (0.00) | 0.33 (0.00) | 0.11 (0.00) | - | 124.6 (0.00) | 140.6 (0.00) |
| | Number of plans elaborated to improve the process | 0.47 (0.00) | 0.31 (0.00) | 0.40 (0.00) | 0.28 (0.00) | 0.28 (0.00) | 0.29 (0.00) | 0.40 (0.00) | 0.44 (0.00) | 0.25 (0.00) | 0.47 (0.00) | 0.34 (0.00) | 0.24 (0.00) | 0.50 (0.00) | - | 185.3 (0.00) |
| | Number of the improved processes | 0.46 (0.00) | 0.30 (0.00) | 0.18 (0.00) | 0.27 (0.00) | 0.19 (0.00) | 0.37 (0.00) | 0.46 (0.00) | 0.44 (0.00) | 0.23 (0.00) | 0.47 (0.00) | 0.47 (0.00) | 0.33 (0.00) | 0.49 (0.00) | 0.60 (0.00) | - |

Source: Own work based on research result.

## 5. Conclusions

On the basis of the conducted research, 15 attributes describing process maturity of the Polish community offices of urban type were identified. Analyzing the research results, it can be stated that elements of process management are used in most of the surveyed entities. At the same time, one can observe that individual attributes are implemented at different levels in the community offices. Moreover, one can observe that:

- Despite the fairly high level of process identification in the researched community offices, these entities rarely had plans to improve all the processes, and actually improved them.
- The researched entities, while monitoring the implementation of processes, mainly focused on their cost. Less frequently monitored parameters for all the processes were: The time of their duration, their quality, and the level of processes' customers satisfaction.
- The respondents of the survey seemed to underestimate the role of managers and employees when implementing the process approach in the community offices.
- The most common deviations occurring during the implementation of processes in the surveyed entities were ad-hoc responses to emerging crises in the execution of processes. And almost 1/3 of the surveyed entities also improvised the execution of processes and exceeded the assumed time for their execution.
- The results of the analysis of stochastic independence (chi-quadrant independence test) between individual attributes allow us to state that the attributes significantly differ from each other. And only the assessment of all of them allows for correct diagnosis of the level of process maturity of the community offices.

On the basis of the conducted research, it was possible to specify factors (attributes) that describe process maturity of the surveyed entities. Thus, we can indicate which aspects managers of the community offices in Poland should pay special attention to, when introducing and extending the use of the process approach in the entities they manage. Moreover, the obtained results may serve as a basis for creating an author's process maturity model aimed at assessing the level of process maturity of the community offices in Poland. In such model, individual entities could be qualified to the process maturity stages on the basis of the assessment of the specific attributes. The first group could include the community offices where there is a lack of process awareness manifested by a lack of employees' knowledge of the processes, and a lack of identified processes. The second group of the so-called "process initiation" would include the entities where there is an awareness of the need to implement process-based solutions, manifested by the fact that some processes of services provided have been identified in these entities; there is a clear division of responsibilities in the execution of basic processes within an organizational unit; not all identified processes are implemented, and there are also numerous cases of improvisation of process execution by employees. The next group could consist of the community offices in which there is a response to emerging crises related to the execution of processes and the awareness that the success of process management implementation depends on the determination of the managers and employees. In the community offices that could belong to the fourth group, the effects of executed processes are measured (due to their costs, time, quality, and customer satisfaction), but only some of them are managed as a whole. The last group could include the entities where processes are managed and constantly improved. As it was mentioned, the above division may form the basis for building a process maturity model for the community offices, but it requires further research related to the attempt to operationalize individual maturity attributes.

A limitation of further discussion of the presented results is the fact that issues of process maturity of the community offices are not often raised in the available literature on the subject. And the research directions presented in the literature are currently focused on the maturity of other areas[11].

---

[11] See e.g., Szumowski and Cyfert (2018); Joshi and Islam (2018).

Therefore, in our opinion, it is justified to further deepen the knowledge on process maturity of the community offices.

**Author Contributions:** Conceptualization, K.K. and M.R.; methodology, K.K.; formal analysis, K.K. and M.R.; investigation, K.K.; resources, K.K. and M.R.; writing—original draft preparation, K.K.; writing—review and editing, K.K. and M.R. visualization, M.R.; supervision, K.K.

**Funding:** This research was funded by the internal financial resources of the Faculty of Economics of University of Warmia and Mazury in Olsztyn.

**Conflicts of Interest:** The authors declare no conflict of interest.

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
