# Peer review of "Attributes of Process Maturity of Public Administration Units in Poland"

_admsci, doi:10.3390/admsci9040084_

Round 1
Reviewer 1 Report
The authors deal with very interesting issue. However, some aspects of this paper need improvements:
research goal is not explicitly defined value added for the discipline not stressed sufficiently literature review does not sufficiently deal with process management in the public sector the response rate is very high - should be explained would be good to annex the questionnaire - the text of questions would allow to check the reliability of responses I do not understand the sense of hypotheses and the sense of the Table 5 - what is the value added of such table???? discussion is missingAuthor Response
First of all, we would like to thank you for your commitment and for your constructive and profound feedback. In summary, we feel that our manuscript was substantially improved by the Reviewer’s comments. We have modified the manuscript accordingly, and detailed corrections are listed below, point by point. We are looking forward to your positive response.
The authors deal with very interesting issue. However, some aspects of this paper need improvements.
We really appreciate Reviewer’s overall positive acceptance of our article.
Research goal is not explicitly defined.
Thank you very much for this valuable notice. Indeed, in our research, we not only identified factors describing process maturity of the community offices of urban type in Poland, but also assessed their interdependence. Therefore, as suggested by the Reviewer, we reformulated the purpose of our study, supplementing it with the aspect of interdependence of factors.
Value added for the discipline not stressed sufficiently.
Thank you very much for this valuable notice. Following the Reviewer's suggestion, we have supplemented our manuscript with the practical application of our research in the management of the community offices of urban type in Poland (in the "Conclusion" section). Bearing in mind that, due to the conducted research, we know what factors influence the process maturity of the surveyed entities, we can indicate which aspects managers of the community offices in Poland should pay special attention to when introducing and extending the use of the process approach in them. This will increase the process maturity of these entities and, as a consequence, increase their efficiency.
Literature review does not sufficiently deal with process management in the public sector.
In accordance with the Reviewer's doubt, which of course we perceive as legitimate and are very thankful for, we have supplemented the literature review with selected current positions on the issue of process management in public organizations.
The response rate is very high - should be explained.
We have obtained such high response rate, because in Poland public information is „open” and public organizations are very disciplined in participating in various research. In addition, the research was commissioned to a company specialized in conducting CATI research.
Would be good to annex the questionnaire - the text of questions would allow to check the reliability of responses.
The results of the presented research constitute a fragment of a broader research on the application of contemporary management concepts in the community offices of urban type in Poland. At the Reviewer's request, we can attach the part of the questionnaire regarding the identification of factors describing the process maturity of these entities.
I do not understand the sense of hypotheses and the sense of the Table 5 - what is the value added of such table?
By reformulating the purpose of our study, as suggested by the Reviewer, in our opinion the legitimacy of making and verifying the hypotheses contained in the manuscript increases (table 5).
Discussion is missing.
Of course we agree with the Reviewer's notice. While preparing our article, we tried to include elements of the discussion in the "Conclusion" section. In response to the Reviewer's valuable notice regarding the refinement of the "Conclusion" part, we have supplemented it with the practical application of our research in the management of the community offices of urban type in Poland.
Reviewer 2 Report
Good morning,
Thank you for allowing me to read your manuscript. I suggest the following:
Process management needs to be succinctly defined in the introduction as it relates to your research. Edit throughout for clarity, organization, syntax. It takes a couple of readings to discern your meaning in places. How the research methodology relates to the model you suggest in your abstract is vague. Do your hypothesis address this proposed model? Describe your intentions with the model more clearly and how the research adds or detracts from it. In your final analysis be specific about what your research contributes to the body of knowledge about process management. What does it add to the theory you employed? How does it help public managers? There are bits and pieces that seem important, but write a stronger conclusion that summarizes better and leads the project forward.
Author Response
Thank you very much for taking your time to evaluate our manuscript and for your constructive and very helpful comments. They allowed us to improve our article. All changes introduced, in accordance with the comments of the Reviewer, are presented below, point by point.
Process management needs to be succinctly defined in the introduction as it relates to your research.
An explanation of the concept of "process management" has been added to the Introduction part.
Edit throughout for clarity, organization, syntax. It takes a couple of readings to discern your meaning in places.
Moderate English changes required.
Thank you very much for this remark. We hope that our review and amendments to our manuscript have allowed it to be sufficiently improved in terms of clarity, organization and syntax. After accepting the amendments, the manuscript will be sent for language correction to improve the quality of the translation.
How the research methodology relates to the model you suggest in your abstract is vague. Do your hypothesis address this proposed model? Describe your intentions with the model more clearly and how the research adds or detracts from it.
The model mentioned in the article was only an inspiration for us to undertake research aimed at identifying factors describing the process maturity of the community offices of urban type in Poland. It was not the basis for constructing the questionnaire (it may be attached at the Reviewer's request). In our opinion, the reason for this issue, and the Reviewer's doubts, are translation-related. During the translation process, the word 'inspiration' was incorrectly rendered. In order not to duplicate this error, the model reference was removed from the abstract.
In your final analysis be specific about what your research contributes to the body of knowledge about process management. What does it add to the theory you employed? How does it help public managers?
Thank you very much for this valuable remark. Following the Reviewer's suggestion, we have supplemented our manuscript with the practical application of our research results in the community offices of urban type in Poland (in the "Conclusion" section).
There are bits and pieces that seem important, but write a stronger conclusion that summarizes better and leads the project forward.
Conclusion part was improved and supplemented with future research directions in the presented scope.
Round 2
Reviewer 1 Report
Despite to some remaining issues (f.e. good discussion is still missing, explanation of high response rate not inserted into the text, statistical significance of sample not explained) the paper can be published - the authors may still try.
Author Response
First of all we would like to thank the Reviewer for the additional effort of reviewing our manuscript. As responding to the second review, which is:
"Despite to some remaining issues (f.e. good discussion is still missing, explanation of high response rate not inserted into the text, statistical significance of sample not explained) the paper can be published - the authors may still try."
we would like to explain that:
- A limitation of further discussion of the presented results is the fact that issues of process maturity of the community offices are not often raised in the available literature on the subject. And the research directions presented in the literature are currently focused e.g. on the maturity of other areas. Therefore, in our opinion, it is justified to further deepen the knowledge on process maturity of the community offices.
We added the comment to our manuscript in "Conclusion" part.
We also supplemented our manuscript with the explanation of the high response rate and we explained that the chosen sample (269 community offices of urban type) accounted for 100% of all the community offices of urban type in Poland, where the municipals had more than 20.000 inhabitants.